# Structural basis for potent neutralization of human respirovirus type 3 by protective single-domain camelid antibodies

Nicole V. Johnson [1,5], Revina C. van Scherpenzeel[2,5], Mark J. G. Bakkers [3,4], Ajit R. Ramamohan [1], Daan van Overveld [3], Lam Le[3], Johannes P. M. Langedijk [3,4], Joost A. Kolkman[2] & Jason S. McLellan [1] ✉

Respirovirus 3 is a leading cause of severe acute respiratory infections in vulnerable human populations. Entry into host cells is facilitated by the attachment glycoprotein and the fusion glycoprotein (F). Because of its crucial role, F represents an attractive therapeutic target. Here, we identify 13 F-directed heavy-chain-only antibody fragments that neutralize recombinant respirovirus 3. High-resolution cryo-EM structures of antibody fragments bound to the prefusion conformation of F reveal three distinct, previously uncharacterized epitopes. All three antibody fragments bind quaternary epitopes on F, suggesting mechanisms for neutralization that may include stabilization of the prefusion conformation. Studies in cotton rats demonstrate the prophylactic efficacy of these antibody fragments in reducing viral load in the lungs and nasal passages. These data highlight the potential of heavy-chain-only antibody fragments as effective interventions against respirovirus 3 infection and identify neutralizing epitopes that can be targeted for therapeutic development.

Paramyxoviruses are a family of enveloped, negative-sense RNA viruses that include several important human pathogens such as human orthorubulavirus 2 and 4 (previously known as human parainfluenza virus (PIV) 2 and 4), human respirovirus 1 and 3 (RV1 and RV3, previously PIV1 and PIV3), measles virus (MeV), mumps virus (MuV), Hendra virus (HeV), and Nipah virus (NiV). RV3 is a common seasonal respiratory virus that infects most children by age three and can cause upper and lower respiratory tract symptoms, including bronchiolitis and pneumonia[1,2]. Illness resulting from RV3 infection is typically more severe in children, accounting for ~29,000 hospitalizations annually in US children under five[3]. Although RV3 infection elicits a neutralizing antibody response, reinfection is common throughout life. Symptoms in adults tend to be mild but can progress into severe and lethal pneumonia in the elderly and immunocompromised individuals[4,5]. Currently, no approved

vaccines are available for RV3 prevention, and no effective antivirals are available for treatment. The substantial disease burden imposed by RV3 infection underscores an urgent need for prophylactic and therapeutic interventions.

Enveloped viruses enter cells through fusion of the viral and host cell membranes. Like other paramyxoviruses, RV3 mediates fusion through a coordinated mechanism requiring two membrane-anchored glycoproteins on the viral surface: the hemagglutinin-neuraminidase protein (HN) and the trimeric fusion glycoprotein (F)[6-8]. HN is responsible for receptor engagement, which triggers F to undergo structural rearrangements that result in membrane fusion[9,10]. F—a class I fusion protein—is initially expressed as an inactive precursor (F0) that requires processing by a host-cell protease into disulfide-linked F1 and F2 subunits to become fusion-competent[11]. Cleavage occurs at a conserved RTKR sequence that can be targeted by TMPRSS2 or other

[1]Department of Molecular Biosciences, The University of Texas at Austin, Austin, TX 78712, USA. [2]Janssen Infectious Diseases and Vaccines, 2340 Beerse, Belgium. [3]Janssen Vaccines & Prevention BV, Leiden, The Netherlands. [4]Present address: ForgeBio B.V., Amsterdam, The Netherlands. [5]These authors contributed equally: Nicole V. Johnson, Revina C. van Scherpenzeel. ✉e-mail: jmclellan@austin.utexas.edu

trypsin-like proteases in the trans-Golgi network or at the plasma membrane[12,13]. The prefusion conformation of F (preF) is a metastable structure composed of a globular head region connected to a helical stalk formed by heptad repeat B (HRB) and a transmembrane domain that extends into the viral membrane (Fig. 1a, Supplementary Fig. 1)[14]. The head region contains three domains (DI-DIII) and two additional HR domains (HRC and HRA) separated by the F1/F2 cleavage site. Cleavage exposes the hydrophobic fusion peptide (FP) at the N-terminus of HRA within F1. After receptor engagement by HN, the HRA domains extend and the FP is inserted into the host cell membrane, forming a pre-hairpin intermediate[8,15]. Subsequent refolding of F into the highly stable postfusion (postF) conformation is driven by interactions between the HRA and HRB helices that collapse

to form a 6-helix bundle characteristic of class I fusion proteins and results in formation of the fusion pore[16,17].

Antibodies that target fusion proteins can inhibit the conformational changes required for fusion, thereby preventing infection. These neutralizing antibodies can be effective interventions for viral infections, and a subset that target class I fusion proteins have been approved for use against SARS-CoV-2, respiratory syncytial virus (RSV), and Ebola virus infection[18–21]. Generally, antibodies that target the prefusion conformation of F are more potently neutralizing than those that bind postF[22]. Immunization of mice with a stabilized RV3 preF antigen resulted in higher neutralizing antibody titers than those elicited with postF, revealing that the prefusion conformation is the primary target for the immune response. Neutralizing antibodies

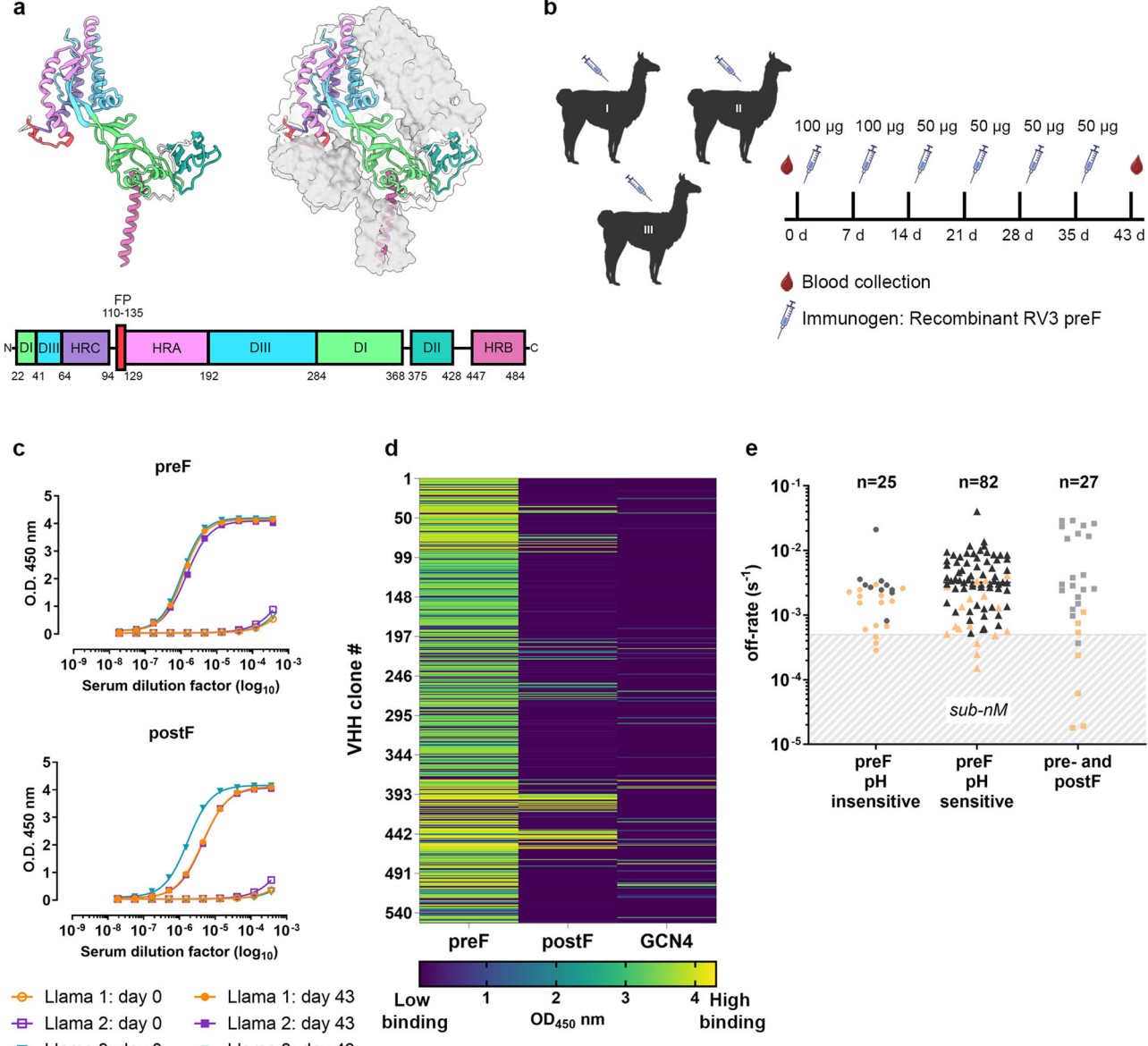

**Fig. 1 | VHHs isolated from llamas immunized with RV3 preF. a** Cartoon representation of the RV3 preF protomer (left) colored according to the linear schematic below, and the trimer (right) with additional protomers shown as gray surface representations. DI–DIII Domains I–III, HRA–HRC Heptad Repeat A–C, FP Fusion peptide. **b** Three llamas were immunized with recombinant RV3 prefusion F (preF) according to the schedule shown. **c** Serum titers from the three immunized llamas before (day 0) and after completion of the immunization protocol (day 43), tested for binding to preF or postfusion F (postF). **d** After two rounds of phage-display

panning, a selection of clones was tested by ELISA for binding to purified RV3 preF, postF, and a GCN4 peptide. Results are shown with a color gradient: high levels of binding indicated by yellow and low binding indicated by navy blue. **e** Off-rate analysis results for a selection of clones that bound preF or both pre- and postF, selected based on the results from **d**. 39 clones, represented as orange points, were selected for further characterization. Source data are provided within the Source Data file. Figure **b** created with BioRender.com released under a Creative Commons Attribution-NonCommercial-NoDerivs 4.0 International license.

isolated from RV3-positive donor sera target two antigenic sites on preF: one at the apex of the trimer and one near the C-terminus of the α2 helix within HRA[14,23,24]. This suggests that antibodies elicited by natural infection may be limited to a few epitopes distal from the viral membrane and illustrates a need to identify diverse alternatives to monoclonal antibodies as potential therapeutics.

Unlike conventional antibodies, which contain two identical heavy and light chains, camelids produce dimeric heavy-chain-only antibodies that have a constant fragment (Fc) attached to a single variable domain (VHH)[25,26]. VHH domains have unique characteristics that highlight their potential in the development of next-generation antibody-based therapeutics such as their small size (~15 kDa) and long complementarity-determining regions (CDRs) that enable targeting of epitopes inaccessible to human or mouse antibodies[27]. These fragments also have advantageous biophysical properties, including higher thermal and chemical stability than most conventional antibodies, which enable long-term storage and simplify transportation[28]. Further, VHHs can be engineered into bivalent VHH-Fc fusions to increase their half-life, enhance target recognition through avidity, and activate Fc-mediated immune functions.[28].

Here, we immunized three llamas with RV3 F to generate phage libraries of F-directed VHH domains. Using phage-display, we identified a panel of VHH candidates that bound to RV3 pre- and/or postF and screened them for neutralization of recombinant RV3-eGFP virus. Candidates with high neutralization potency were sorted into epitope bins using biolayer interferometry competition assays. We then determined high-resolution cryo-EM structures of potently neutralizing VHHs with non-overlapping epitopes complexed with RV3 preF to investigate the molecular basis for epitope recognition and neutralization. VHH-Fc fusion constructs were used to prophylactically treat cotton rats before challenge with RV3 to determine their efficacy in reducing viral load in the nose and lungs. Our findings provide important insight into neutralizing epitopes that can be targeted by camelid VHHs and implications for their development into effective treatments to combat respiratory infections.

## Results

### Isolation of camelid VHHs that target RV3 F
To generate VHH domains that selectively bind to the prefusion conformation of RV3 F, three llamas were immunized with recombinant prefusion RV3 F antigen containing a C-terminal GCN4 trimerization domain and one stabilizing substitution in the negatively charged, repulsive cluster above HRB, similar to a stabilizing substitution (D452N) in RSV F (Supplementary Fig. 2)[29]. Different doses were administered by subcutaneous injection every 7 days over a total of 35 days (Fig. 1b). Blood was drawn on day 0 and day 43 post-immunization. Plasma prepared after 43 days was screened by ELISA for serum titers against RV3 pre- and postF to determine a positive overall immune response (Fig. 1c). Two consecutive rounds of panning were performed under different selection conditions to isolate high-affinity, RV3 F-targeting VHHs from the constructed phage display libraries. In the first round, all phage libraries were panned against biotinylated RV3 preF, with or without counterselection for RV3 postF and GCN4 peptide. Counterselection with RV3 postF and GCN4 peptide was included to eliminate postF and GCN4 binders and enrich for preF-specific VHHs. In the second round, the enriched phage outputs from round one were incubated with lower concentrations of preF for increased stringency. Two rounds of iterative panning on immobilized preF protein with counterselection using postF and GCN4 resulted in a 15,000-fold enrichment of possible candidate preF-binders. Subsequently, over 500 clones were randomly picked and tested as periplasmic extracts for binding in ELISA to RV3 preF, RV3 postF, and GCN4 peptide (Fig. 1d). From this, we identified a total of 332 clones demonstrating specificity for preF, and 108 clones exhibiting binding to both pre- and postF. Additionally, three clones were identified as

postF-specific, with an additional 43 clones displaying affinity toward the GCN4 peptide. Clones that exhibited preF-specificity or bound to both pre- and postF protein were sequenced and representative members from each CDR3 family (137 clones total) were subjected to further evaluation for their interactions with RV3 F proteins based on off-rate analysis by surface plasmon resonance (SPR) (Fig. 1e). Biotinylated pre- or postF antigens were immobilized on a streptavidin-coated sensor chip, followed by flowing diluted VHH-containing periplasmic extracts over the coated surface. Within this cohort, 51 clones were confirmed as preF-specific, while 25 clones exhibited binding to both pre- and postF antigens. Dissociation rate constants were observed ranging from ~$3 \times 10^{-2}$ to $2 \times 10^{-5}\,s^{-1}$. Most of the assessed VHHs demonstrated weak affinities for F, evidenced by off-rate constants greater than $5 \times 10^{-3}\,s^{-1}$. Among the 137 clones initially assessed, 87 clones bound F by ELISA but exhibited no association to pre- or postF upon SPR analysis. This discrepancy led us to speculate that exposure to extreme pH during the ligand regeneration SPR cycles (10 mM glycine, pH 2.5 followed by 10 mM NaOH/1 M NaCl, pH ~11) might induce conformational changes in F, affecting a subset of epitopes present at neutral pH. To evaluate these clones, we conducted SPR experiments using a Biotin CAPture reagent immobilized onto a CAP sensor chip, which required F immobilization between each VHH measurement and maintained the F antigens in a neutral pH solution. These experiments identified an additional 82 clones displaying preF specificity and two preF- and postF-binding VHHs with off-rate constants similar to those observed in the prior experiments (Fig. 1e). Two of these clones did not bind F. In total, we successfully identified 107 preF-specific and 27 conformation-agnostic RV3 F-directed VHHs. Clones that bound preF with slow off-rates (less than $5 \times 10^{-3}\,s^{-1}$) were assigned to a VHH cluster based on CDR3 sequence homology. The strongest binder from each cluster was selected to generate a panel of 39 lead VHH candidates to recombinantly express and purify for further characterization.

### Neutralizing VHHs target three antigenic sites
To evaluate the antiviral activity of our lead VHHs, we performed neutralization assays using a recombinant RV3 virus that expresses GFP (RV3-eGFP), enabling fluorescent detection of viral replication in cells (Fig. 2a, Supplementary Fig. 3a)[30]. Of the 39 VHHs tested, 16 were able to neutralize RV3 in this assay, with eight VHHs displaying half-maximal inhibitory concentration ($IC_{50}$) values in the low to sub-nanomolar range (Fig. 2a, Supplementary Tables 1 and 2). Of these, VHHs 5E03, 6B03, and 5D03 were excluded from further analysis after small precipitates were visually observed in the protein samples. None of the tested VHHs neutralized recombinant RV1-eGFP or PIV2-eGFP viruses (Supplementary Fig. 3b, c), indicating that their epitopes are not strictly conserved among closely related paramyxoviruses. Binding of the 13 remaining VHHs to RV3 preF was confirmed by ELISA (Fig. 2b). Among the subset of eight VHHs displaying potent neutralization, two clones, namely 1D10 and 3B04, also bound postF, with 50% effective concentration ($EC_{50}$) values of 32 and 26 nM, respectively (Fig. 2b, Supplementary Table 3). This observation indicates that at least one neutralization-sensitive antigenic site is maintained in the postfusion conformation.

Although several antibodies have been isolated that target the RV3 F protein, the antigenic landscape of RV3 F remains incompletely characterized[14,23,24,31,32]. To gain more insight into the epitopes targeted by our high-affinity, neutralizing VHHs, we conducted a series of competition binding assays via biolayer interferometry to categorize these VHHs into distinct epitope bins (Supplementary Fig. 4a). Streptavidin biosensors were used to capture biotinylated RV3 preF before being dipped into wells containing 1 of the 13 VHHs to achieve saturation. Following a short dissociation phase, the RV3 F−VHH complex was dipped into a well containing the same VHH, a different VHH, or a buffer solution. Competition between VHHs was defined by

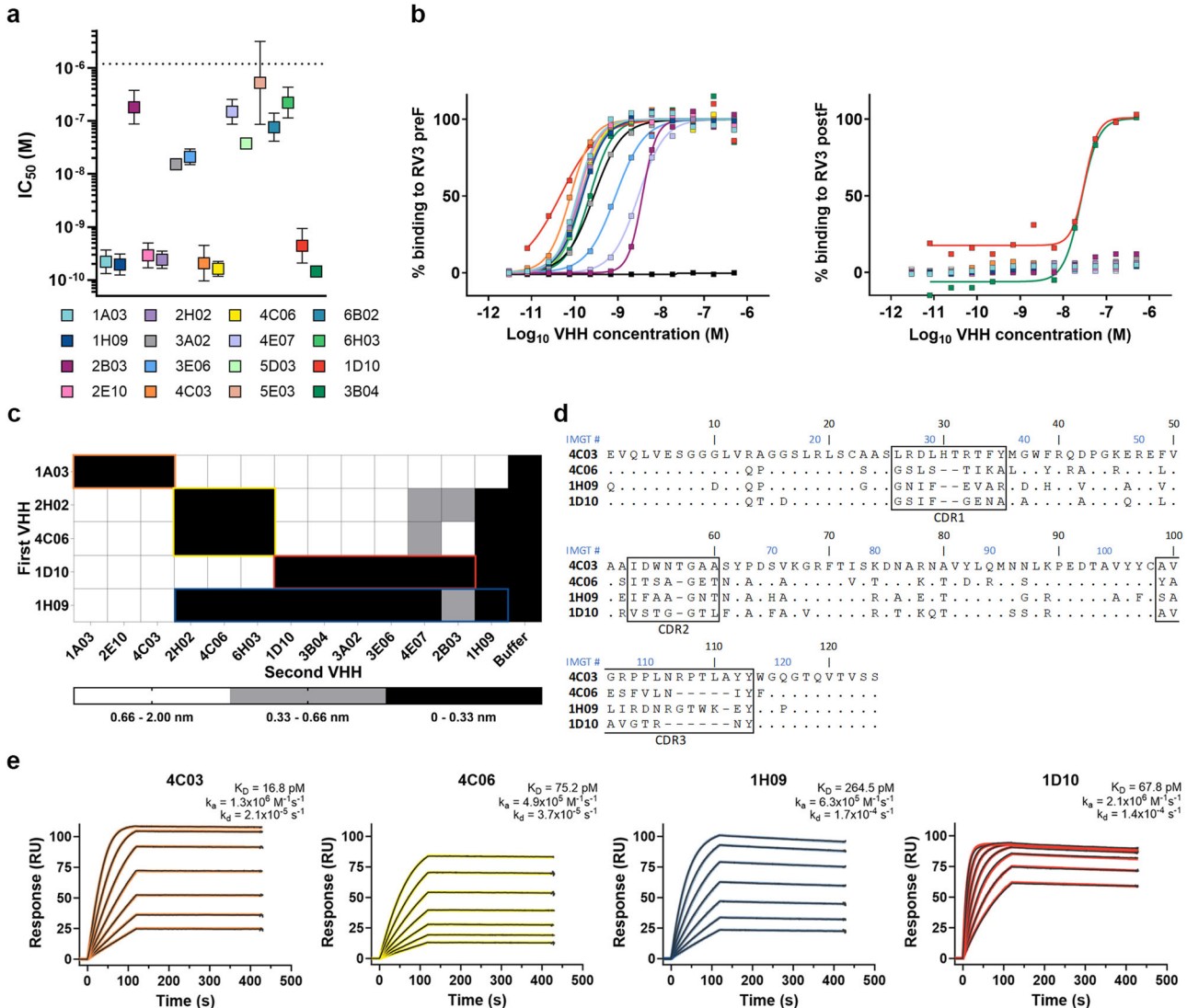

**Fig. 2 | Potently neutralizing VHHs bind to three non-overlapping antigenic sites. a** Half-maximal inhibitory concentration (IC$_{50}$) values for 16 potently neutralizing VHH candidates determined by neutralization assays performed with recombinant RV3-eGFP virus. Points represent geometric means of biological replicates ($n = 2$ for 2B03, 3A02, 3E06, 4E07, 5D03, 5E03, 6B02, and 6B03; $n = 3$ for 1A03, 1H09, 2E10, 2H02, 4C03, and 4C06; $n = 4$ for 1D10 and 3B04) ±geometric standard deviations. Representative curves are displayed in Supplementary Fig. 3 and numeric IC$_{50}$ values are listed in Supplementary Table 1. **b** ELISA binding curves for the top 13 VHH candidates. VHHs were tested for binding to RV3 preF (left) and postF (right). Half maximal effective concentration (EC$_{50}$) values are listed in Supplementary Table 3. **c** Competition matrix indicating the level of

binding observed for each of the 13 top VHH candidates to preF saturated with the same VHH or a different VHH from the same pool. Columns indicate the competition profile of each VHH. Each epitope bin is outlined based on the representative VHH that was selected for further characterization, colored as in **a** and **b**. **d** Sequence alignment of the representative VHH selected from each epitope bin. VHHs 4C06, 1H09, and 1D10 are aligned to the 4C03 sequence. IMGT-based residue numbers are represented by blue text. Invariant residues are indicated by black dots. Complementarity determining regions (CDRs) are indicated by boxes. **e** Surface plasmon resonance sensorgrams for binding of each representative VHH to preF. Binding curves are colored black. Data fit to a 1:1 binding model is colored by VHH according to **a** and **b**. Source data are provided within the Source Data file.

the level of response (nm) observed for binding of the second VHH. VHHs that competed for binding to an overlapping epitope (0–0.33 nm) were assigned to the same epitope bin, while those that bound strongly to the complex (0.66–2 nm) were assigned to a separate bin. Intermediate response values (0.33–0.66 nm) can indicate minor epitope overlap or exposure of a shared epitope through partial dissociation of the first VHH and were not considered for bin assignment. These experiments identified three apparent epitope bins (Fig. 2c). The largest bin (bin 1) comprises 6 of the 13 VHHs tested: 1D10, 2B03, 3A02, 3B04, 3E06, and 4E07. Notably, the two pre/postF binding VHHs, 1D10 and 3B04, are included in bin 1. The two remaining bins each comprise three VHHs: bin 2 includes 2H02, 4C06, and 6H03, whereas bin 3 contains VHHs 1A03, 4C03, and 2E10. Interestingly, VHH 1H09 displayed competition with members from bins 1

and 2. Based on these data, we selected one VHH from each bin (1D10, 4C06, 4C03, and bridging VHH 1H09) for more in-depth characterization (Fig. 2d).

To better understand the binding kinetics of the selected VHHs, we determined the equilibrium dissociation constants ($K_D$) by SPR using the CAP sensor chip previously employed in our larger screen (Fig. 2e). VHHs 4C03 and 4C06, which bind non-overlapping preF-specific epitopes, bound preF with affinities of 16.8 pM and 75.2 pM, respectively. 1H09, which binds an epitope that bridges two of our identified epitope bins, bound preF with a lower affinity of 264.5 pM due to a combination of a slower on-rate constant ($k_a = 6.3 \times 10^5 \, M^{-1}s^{-1}$) and faster off-rate constant ($k_d = 1.7 \times 10^{-4} \, M^{-1}s^{-1}$). 1D10, which also binds postF, bound to preF with an affinity similar to 4C03 and 4C06, with a $K_D$ of 67.8 pM. We also attempted to determine the binding

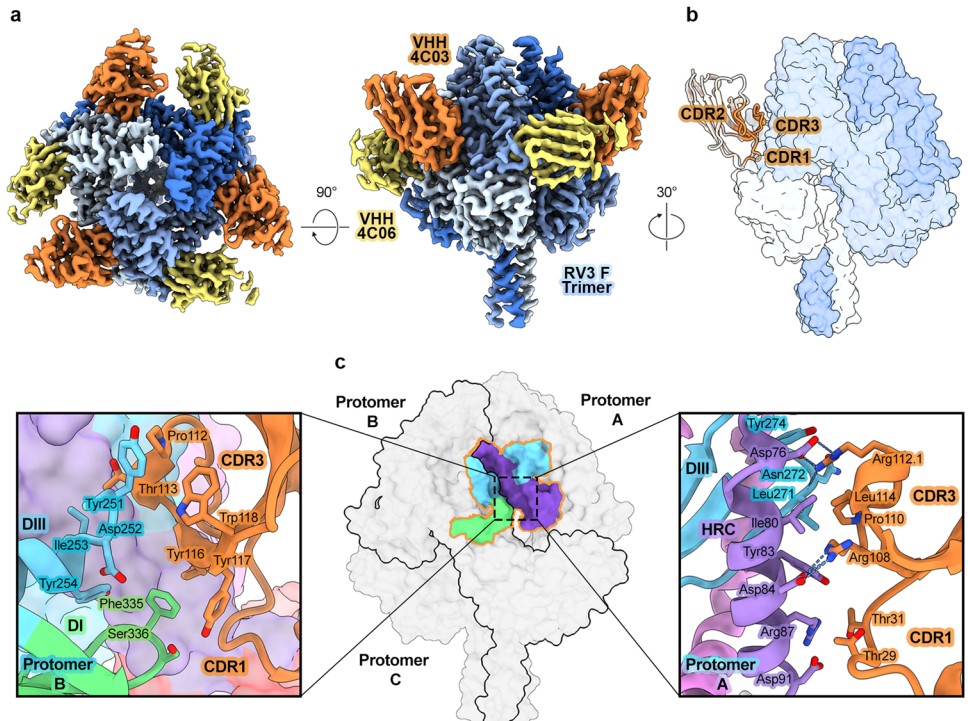

**Fig. 3 | VHH 4C03 binds a quaternary epitope on preF centered between two adjacent protomers. a** Cryo-EM map of 4C03 and 4C06 bound to preF shown in top-down and side views. 4C03 is colored orange, 4C06 is colored yellow, and each protomer of preF is colored a shade of blue. **b** Side view of preF with one bound 4C03 VHH shown. The preF trimer is shown as partially transparent surface, colored as in **a**. 4C03 is shown in light peach with the CDRs colored orange. **c** (center) The quaternary epitope of 4C03 outlined on a surface representation of preF. PreF is colored gray and the epitope footprint is outlined in orange. Residues within the epitope footprint are colored according to domain within preF as in

Fig. 1a. DI is colored light green, DIII is colored light blue, HRC is colored purple. (left) Zoomed-in view of the 4C03 interface with protomer B, shown as cartoons. Select regions of the model have been omitted for clarity. Important residue contacts are shown as sticks with oxygen atoms colored red and nitrogen atoms colored blue. The adjacent protomer (protomer A) is shown as transparent surface, colored by domain. (right) Zoomed-in view of the 4C03 interface with protomer A, shown as cartoons. Select regions of the model have been omitted for clarity. Important residue contacts are shown as in (left), with hydrogen bonds and salt bridges shown as blue dashes.

affinities of the four VHHs for postF, however, the response curves in these experiments were too low to determine reliable kinetic values (Supplementary Fig. 4b–e).

### 4C03 and 4C06 bind distinct epitopes on prefusion F

To investigate the molecular basis for neutralization-sensitive epitopes targeted by preF-specific VHHs, we determined a 2.6 Å resolution cryo-EM structure of preF bound to VHHs 4C03 and 4C06 (Fig. 3a, Supplementary Table 4). Both VHHs bind previously undescribed epitopes around the equator of the globular head domain of preF.

4C03 binds a quaternary epitope, with 442.5 Å² of buried surface area (BSA) on the first protomer (protomer A) and 380.3 Å² BSA on the adjacent protomer (protomer B) (Fig. 3b). The VHH approaches the F trimer at a steep angle, leading to an expansive interface along the HRC helix of protomer A and substantial contacts with DI and DIII of protomer B (Fig. 3c). The 4C03 binding interface is dominated by contacts within CDR3, although CDR1 forms a modest interface at the cleft between the F protomers. CDR2 is excluded from the interface, with the closest residues 7.4 Å apart, as measured from their respective terminal sidechain atoms (Fig. 3b). CDR1 bridges the protomers through nonpolar contacts near the base of the HRC helix of protomer A and the β11-β12 loop in DI of protomer B (Fig. 3c: left, right). 4C03 contains the longest CDR3 of the 4 VHHs, with 15 residues forming an extended loop that curls to lie across the epitope while accommodating the steep angle of approach. On protomer A, CDR3 forms interactions that span the HRC helix and the proximal β5-β6 hairpin within DIII. Asp76 and Asp84 within HRC form salt bridges with Arg112.1$_{CDR3}$ and Arg108$_{CDR3}$, respectively, flanking the interacting

region (Fig. 3c, right). The CDR3 loop also contacts DIII on protomer B residues Tyr251–Tyr254, upstream of the α7 helix and joins CDR1 to bury Phe335 and Ser336 within DI (Fig. 3c, left). Structural rearrangements during the transition from the pre- to postfusion conformation break the interprotomer interface at the center of the 4C03 quaternary epitope, explaining its specificity for preF. Further, the nearly even distribution of surface area buried by 4C03 on each protomer suggests that binding prevents this separation and locks F in the prefusion state, providing the basis for neutralization.

4C06 forms a large interface, burying 1227 Å² of surface area on the preF trimer. Its epitope lies almost entirely within a single protomer of preF (1094 Å² BSA), with minor contacts on neighboring loops of DII on the adjacent protomer (133 Å²) (Fig. 4a). 4C06 bridges multiple domains of the F protomer, with CDR1 and CDR3 tightly packed into the pocket formed between the FP, the α3 helix within HRA, and the two-strand anti-parallel β-sheet (β1 and β6) within DIII. The interweaving nature of the interaction involves residues 80–85 within framework region 3 (FR3), which is sometimes referred to as a fourth CDR[33]. CDR1, CDR2, and the FR3 loop surround the FP at the F1/F2 protease recognition site between Arg109 and Phe110 (Fig. 4b, top). Here, the CDR1 loop tucks between the FP and the α3 HRA helix. Within the adjacent FR3 loop, Asn85 forms two hydrogen bonds with the backbone atoms of Arg109 and Phe111, spanning the cleavage site. CDR2 joins the other two loops to form a deep cavity that buries the sidechain of Phe111. On the opposite side of the CDR1 loop, CDR3 reaches between the α3 HRA helix and β1 of DIII, connecting the domains through a web of seven hydrogen bonds (Fig. 4b, bottom). This network of interactions mediated by CDR3 likely acts as a

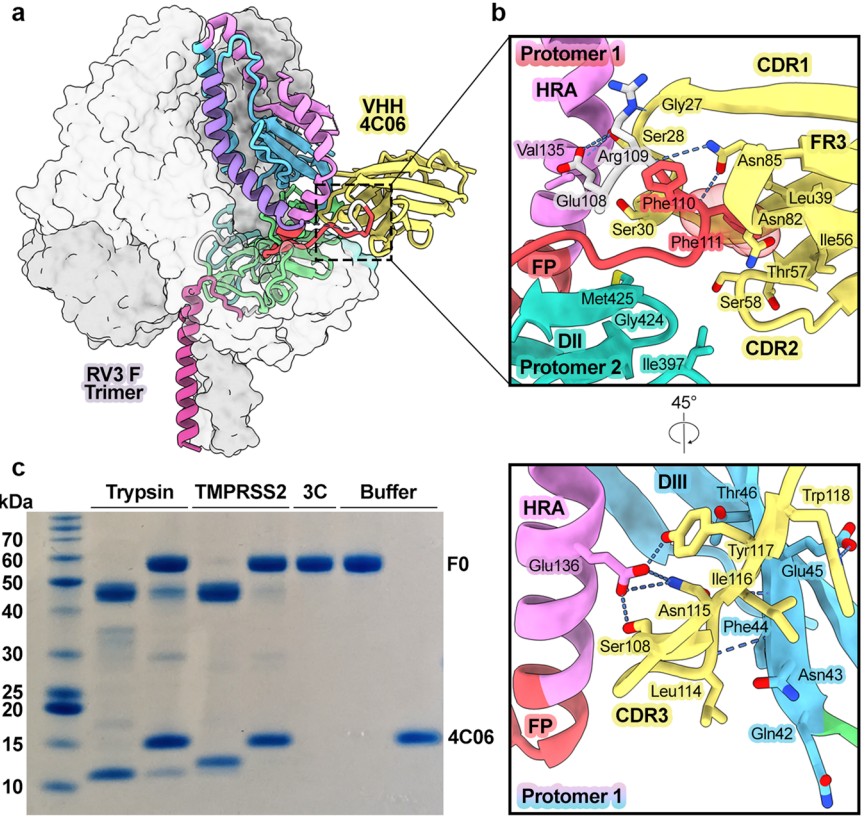

**Fig. 4 | VHH 4C06 binds preF between the fusion peptide and heptad repeat A. a** Side view of preF with one bound 4C06 VHH shown. One protomer is shown as cartoons, colored according to domain as in Fig. 1a. The remaining protomers are represented as partially transparent, gray surfaces. 4C06 is shown as a yellow cartoon. **b** Zoomed-in views of the 4C06 interface with preF rotated by 45˚ about the trimer axis. Irrelevant sections of preF and 4C06 have been omitted for clarity. Important residues are shown as sticks. Oxygen atoms are colored red, nitrogen atoms are colored blue, hydrogen bonds and salt bridges are shown as blue dashes. **c** Reducing SDS-PAGE gel showing cleavage of preF (F0) by trypsin or TMPRSS2 in the presence or absence of 4C06, or 3C protease (negative control). Gel shown is one of three experiments, with enzyme concentrations optimized for specific cleavage of preF at the F1/F2 site. Source data are provided within the Source Data file.

molecular tape, preventing the extension of HRA away from DIII and inhibiting formation of the pre-hairpin intermediate.

The interaction observed between 4C06 and the F1/F2 cleavage site prompted us to test whether 4C06 binding can prevent proteolytic activation of F0 by known activating proteases. Therefore, we performed a cleavage assay using trypsin or TMPRSS2 to cleave prefusion F0 in the presence or absence of 4C06 (Fig. 4c)[34]. Without 4C06, 100 nM trypsin or TMPRSS2 effectively processed 3.6 μM F0 into the F1 and F2 subunits within 30 min. When F0 was pre-incubated with 4C06, trypsin cleavage was reduced, whereas cleavage by TMPRSS2 was nearly abolished. These results demonstrate that 4C06 binding can inhibit proteolytic cleavage of RV3 F in vitro.

## 1D10 binds domain I of prefusion F

To investigate the neutralizing epitope targeted by the conformation-agnostic VHH 1D10, we determined a 3.4 Å structure of 1D10 bound to preF (Fig. 5a, Supplementary Table 4). 4C06 was included in the complex to add size to the particles and diversify spatial orientation on the EM grid. The structure reveals that 1D10 binds a quaternary epitope at the base of the globular head region near the HRB stalk (Fig. 5b). 1D10 curves tightly around one protomer, burying 718 Å² of surface area using all three CDRs and additional residues throughout the 5-strand β-sheet of the Ig fold framework (Fig. 5b, right). The interaction includes a network of seven hydrogen bonds and two salt bridges formed by residues within CDRs 2 and 3 and residues Arg50 and Arg55 within FR2 (Fig. 5b, right). The interface is centered on the β10-β11

hairpin, which is contacted by residues within CDR1 and CDR3. CDR3 reaches along the β hairpin toward the central cavity of the trimer, contacting a short interprotomer interface formed between DI residues Asp343–Val347 and the DI/DII linker of the adjacent protomer (Fig. 5c, left). 1D10 buries 246 Å² on the second protomer, primarily through interactions mediated by the broad FR1/CDR1 recognition loop, contacting the short β8-β9 hairpin within DI and proximal DII residues (Fig. 5c, left). Residues within the 1D10 footprint have been identified as important for the fusion-regulating interactions of PIV5 F and MeV F with their respective attachment proteins[35–38]. Our structure indicates a possible mechanism by which 1D10 binding disrupts the interaction between preF and HN on the viral membrane.

Since 1D10 can bind to F in its prefusion and postfusion conformations, we aligned our structure to the previously determined model of the RV3 postF trimer (PDBID: 1ZTM) to visualize the putative epitope[16] (Fig. 5c). In the transition from pre- to postfusion, the tertiary structures of DI and DII are conserved, though their spatial arrangement is twisted as the HRB helix collapses to form the resulting 6-helix bundle. The 1D10 epitope within DI of a single protomer retains its structure entirely in the postfusion conformation. However, within trimeric postF, the unstructured loop that comprises the DI/DII linker between β13₀ and β13 on the adjacent protomer clashes with the CDR3 loop (Fig. 5d). Our modeling indicates that a ~4 Å outward shift of DII and the preceding loop would eliminate clashing and is supported by formation of the primary CDR3 interaction at the β10-β11 hairpin immediately proximal to this region on the first protomer.

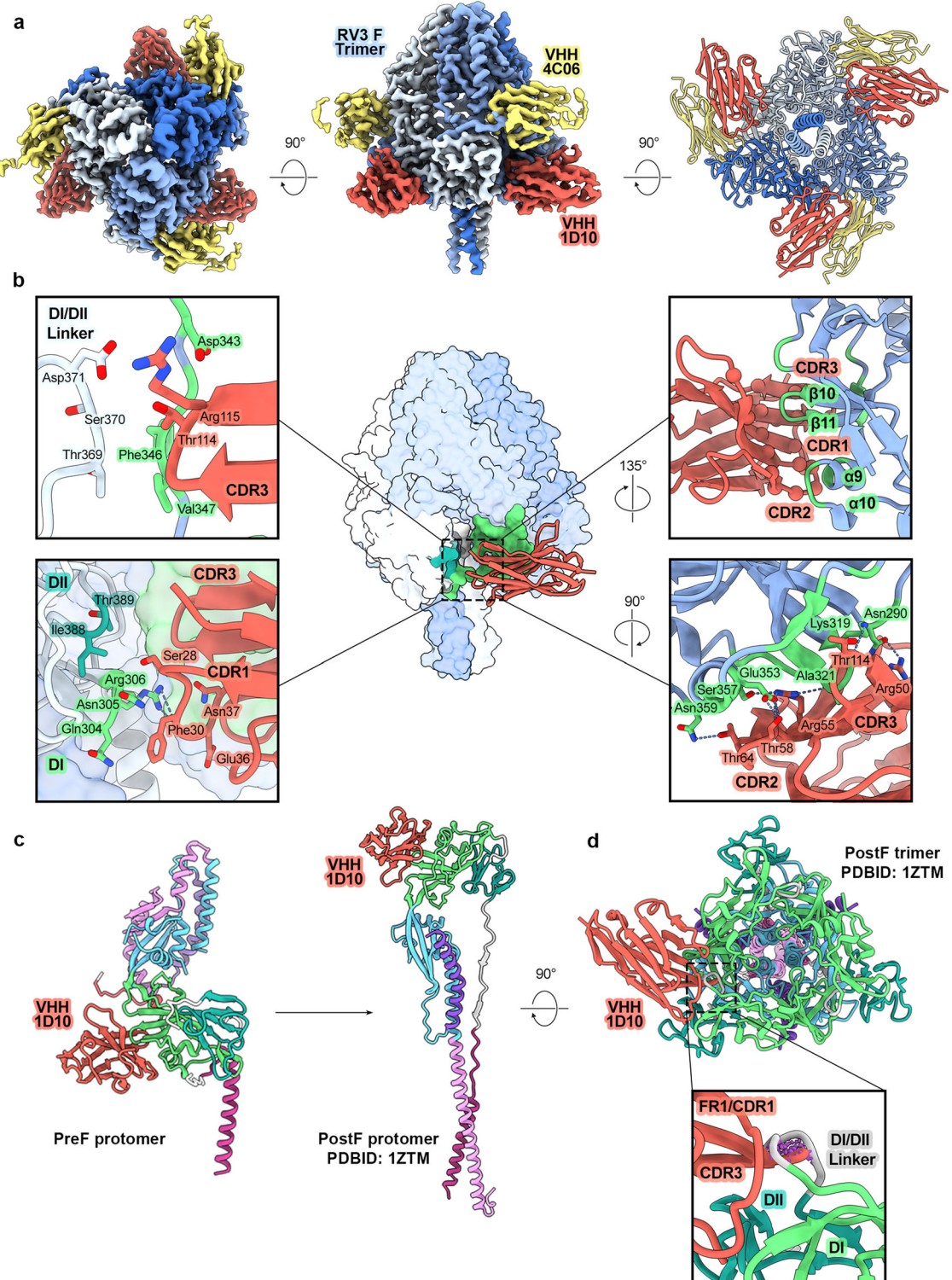

**Fig. 5 | VHH 1D10 binds preF at an epitope near the helical stalk, primarily within DI. a** Cryo-EM map of 1D10 and 4C06 bound to preF shown in top-down and side views, structure shown in bottom-down view. 1D10 is colored red, 4C06 is colored yellow, and each protomer of preF is colored a shade of blue. **b** (center) The preF trimer is represented as light colored surface, with each protomer a different shade of blue. The 1D10 epitope is opaque and colored according to domain as in Fig. 1a. DI is colored green and DII is colored dark turquoise. Insets are zoomed-in views of the interface between 1D10 and preF. (left, top) A small interprotomer interface at the central cavity of preF is contacted by CDR3 of 1D10. Oxygen atoms are colored red, nitrogen atoms are colored blue, hydrogen bonds and salt bridges are shown as blue dashes. (left, bottom) Important residues involved at the

interface between 1D10 and DI and DII of one protomer, shown as sticks. (right, top) The entire concave surface of 1D10 participates in contacts with DI. 1D10 contact residues here are shown as spheres. (right, bottom) 1D10 forms a large network of hydrogen bonds and salt bridges throughout DI of one protomer. **c** (left) 1D10 (red) bound to a single preF protomer, shown as cartoons colored by domain. (right) Model of 1D10 bound to a postF protomer (PDBID: 1ZTM). Modeling and calculations were performed in ChimeraX. DI of the preF protomer bound to 1D10 was aligned to DI of the postF protomer. RMSD: 1.6 Å. **d** Model of 1D10 bound to the postF trimer. Zoomed-in view shows the interface between 1D10 and postF, showing clashes within the DI/DII linker. Clashes are shown as purple dashes, defined by atomic distances of <0.6 Å.

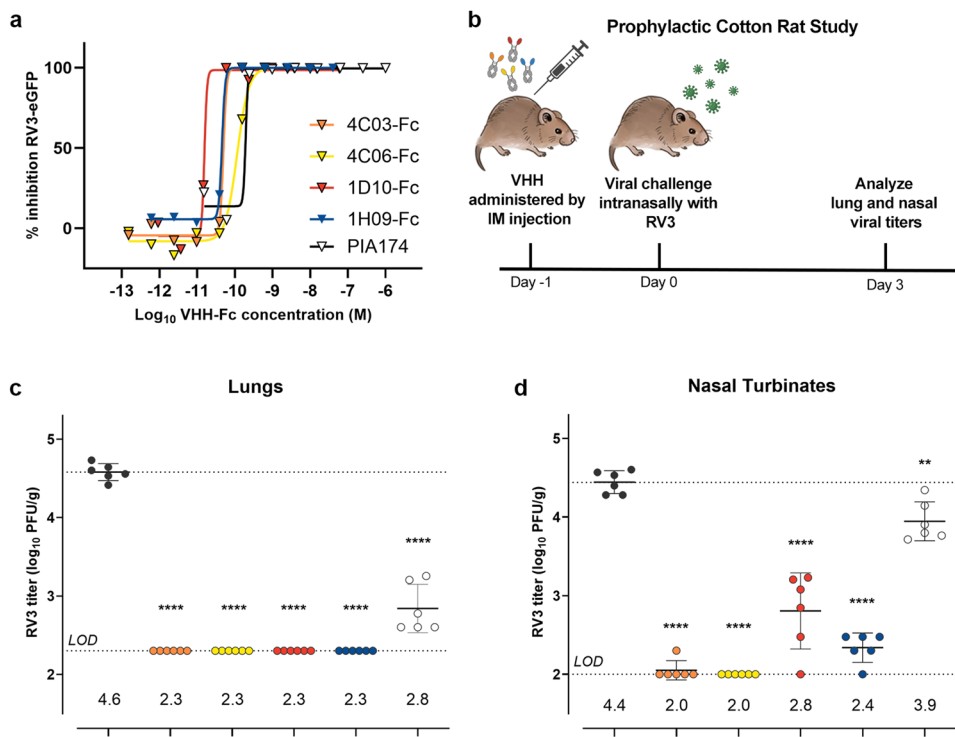

**Fig. 6 | Prophylactic treatment with neutralizing VHH-Fc fusions reduces viral replication in the lungs and nasal passages of cotton rats. a** Representative inhibition curves for each VHH-Fc construct and neutralizing IgG antibody PIA174, resulting from neutralization assays using recombinant RV3-eGFP virus. Calculated $IC_{50}$ values are reported in Supplementary Fig. 5b. **b** Protocol schematic for the prophylactic cotton rat study. Viral titers in the lung (**c**) or nasal turbinates (**d**) of cotton rats treated with either a VHH-Fc, buffer solution (vehicle), or antibody PIA174 (neutralizing anti-RV3 IgG antibody) 2 days after intranasal RV3 challenge, determined by plaque assay. Error bars indicate mean values ± SD ($n = 6$). Dotted line indicates the lower limit of detection (LOD). All groups for both lung and nasal viral titers showed statistically significant difference ($\alpha = 0.05$) compared to vehicle, based on ordinary one-way ANOVA with Dunnett's multiplicity correction. ****$p < 0.0001$, **$p = 0.0067$. Source data are provided within the Source Data file. Figure **b** created with BioRender.com released under a Creative Commons Attribution-NonCommercial-NoDerivs 4.0 International license.

An additional energy requirement to compensate for clashing agrees with our binding data that show 1D10 binds ~1000-fold weaker to postF than preF (Fig. 2b, Supplementary Table 3).

## VHH-Fc fusions reduce replication of RV3 in vivo

To assess the potential of these VHHs as a prophylactic intervention, we used the cotton rat model for protection studies. Cotton rats are susceptible to productive RV3 infection and develop pulmonary disease that closely resembles that in humans[39]. For these studies, each VHH was engineered and expressed as a single-chain human Fc fusion (VHH-hinge-CH2-CH3) for direct comparison to PIA174, a neutralizing F-directed monoclonal antibody (Supplementary Fig. 5a, Supplementary Table 5)[14]. Neutralization assays were performed as previously described, revealing sub-nanomolar potency for each VHH-Fc fusion protein (Fig. 6a, Supplementary Fig. 5b). Groups of six cotton rats were intramuscularly administered either a buffer solution (vehicle control), or a 15 mg/kg dose of 4C03-Fc, 4C06-Fc, 1D10-Fc, 1H09-Fc, or reference antibody PIA174 24 hours prior to a challenge with RV3 (Fig. 6b). An additional group of three rats was dosed with PBS instead of virus as an uninfected control.

At 3 days post-infection, plaque assays were performed to determine the lung viral titers (Fig. 6c). Remarkably, no virus was found above the limit of detection (2.3 $\log_{10}$ PFU/g) in the lungs of groups that were treated with any of the four VHH-Fc proteins, while the cotton rats administered with vehicle had an average titer of 4.6 $\log_{10}$ PFU/g. Interestingly, the VHH-Fc therapies outperformed PIA174 at this dose, which resulted in average titers of 2.8 $\log_{10}$ PFU/g. Nasal viral titers were also assessed at 3 days post-infection (Fig. 6d). The group that received PIA174 exhibited a slight reduction in viral load in the

nose relative to the vehicle group, with an average of 3.9 $\log_{10}$ PFU/g, consistent with observations that systemically delivered IgG antibodies are inefficient at preventing viral replication in these tissues[40]. In contrast, groups treated with either 4C03-Fc or 4C06-Fc had no detectable virus in their nasal passages. The administration of 1D10-Fc or 1H09-Fc led to a more modest reduction in virus titers, measuring 2.8 and 2.4 $\log_{10}$ PFU/g, respectively. Taken together, these data show a robust prophylactic antiviral response by VHH-Fc fusions.

## Discussion

Human respiroviruses present an ongoing threat to public health, accentuated by the absence of approved interventions to treat disease. Antibody-based therapeutics have effectively treated influenza, RSV, Ebola, and SARS-CoV-2 infections[18–21]. However, few antibodies that target RV3 F have been thoroughly characterized. PIA174, a neutralizing antibody isolated from human sera, binds a quaternary epitope at the apex of preF[14]. Recently, four additional neutralizing human antibodies were isolated that compete with PIA174 for binding to preF, indicating this antigenic site—structurally similar to site Ø in RSV F and human metapneumovirus F—is likely dominant in the human repertoire[23,41]. Two antibodies have been described that target a second antigenic site (site X) located slightly lower on the globular head domain of F, within HRA and DIII[23,24]. Here, we identified a panel of neutralizing camelid VHHs that target three distinct antigenic sites on RV3 preF. Our structural data indicate that these VHHs do not bind within sites Ø or X and instead bind three previously uncharacterized neutralizing epitopes.

Antibodies that target RV3 F may conceivably neutralize infection by three main mechanisms: stabilization of a prefusion or intermediate

conformation, inhibition of proteolytic cleavage, and disruption of the interaction between F and HN. We determined that 4C03 binds a quaternary epitope dominated by interactions with a long CDR3 that bridges the HRC helix of one protomer to the junction between DI and DIII of the adjacent protomer. This defines the structural basis for prefusion specificity and suggests that 4C03 may neutralize RV3 by stabilizing the prefusion trimer, though direct experimental evidence is required to confirm this mechanism. Antibody-mediated prefusion stabilization contributes to the neutralization mechanisms of antibodies AM14 and 5B3, which target quaternary epitopes on RSV preF and NiV preF, respectively[42,43]. Prefusion stabilization of class I fusion proteins can similarly be achieved by designing amino acid substitutions that strengthen the interprotomer interface in regions that must move apart during the transition to postfusion[44].

4C06 also binds a quaternary epitope, though the main interaction is within a single protomer. The interaction centers on the HRA helix, extending on one side through extensive contacts with the upstream F1/F2 cleavage site and FP. Within RV3 F—and several other class I fusion proteins—the FP is flexible and partially buried within the trimer[14,45–47]. However, this region is exposed enough to be accessed by neutralizing antibodies[48–51]. Direct binding to the FP indicates that 4C06 neutralizes RV3 by preventing insertion of the FP into the host cell membrane, which is an essential early step in the fusion process. Refolding of F is also likely blocked by binding of CDR3 between the HRA helix and DIII of the adjacent protomer, which must separate by ~100 Å in the transition to postfusion. Further, we showed that binding of 4C06 to uncleaved preF blocks proteolytic cleavage by trypsin and TMPRSS2. Cleavage inhibition has been observed as a neutralization mechanism for antibodies that target other class I fusion proteins, for instance, mAb100 neutralizes Ebola virus by blocking cleavage of its glycoprotein by cathepsins, a crucial step for cellular entry[51]. Similarly, antibodies CR8020 and MEDI8852 inhibit cleavage activation of influenza A hemagglutinin by trypsin-like proteases on the viral surface, thus preventing fusion[52,53]. Blocking the cleavage formation of F1/F2 required for RV3 entry may also contribute to the neutralization activity of 4C06, though it is unknown to what extent uncleaved preF is incorporated in the budding virus.

1D10 is one of two VHHs we determined to bind a neutralization-sensitive antigenic site present in both pre- and postF. Our ELISA data indicates that 1D10 binds to postF with lower affinity, which is supported by our modeling that shows a clash with the adjacent protomer when 1D10 binds the postfusion trimer. Interactions between 1D10 and preF are almost entirely within a single protomer, though binding could result in some stabilization of the preF conformation to promote neutralization. The clashing we observed in our model suggests another possible mechanism for neutralization whereby binding to preF obstructs the inward movement of the domains during the transition to postF and prevents full conversion. However, class I fusion proteins such as RSV F and SARS-CoV-2 S have been shown to adopt conformations with various degrees of trimer opening or "breathing" that could possibly overcome this obstruction[54,55]. Additionally, 1D10-mediated neutralization could involve disruption of the interaction between preF and HN. The triggering mechanism for RV3 F and other paramyxovirus F proteins has been attributed to interactions between the fusion protein and the stalk of the attachment protein[56–58]. Within the fusion protein, these interactions have been localized to the region buried on preF by 1D10 at the interprotomer interface between the Ig-like fold of DII and DI of the adjacent protomer[35–38]. Specific residue substitutions have been identified that inhibit this interaction and decrease fusion activity for PIV5 and MeV, with some overlap between them[37,38]. Indeed, a subset of residues within the 1D10 epitope of RV3 F also align to affected residues in PIV5 or MeV F. Although point mutations within RV3 F have not been examined for their impact on the HN stalk interaction or fusion, 1D10 should be a useful reagent for probing the F-HN interactions.

The small size and stability of VHHs have led to their development as alternatives to conventional antibodies[27]. However, VHHs themselves are limited in their therapeutic use because they lack the Fc region, which is important for serum half-life and activation of cell or complement-based immune functions. Fusion of the VHH domain to appropriate Fc regions overcomes this limitation while maintaining the specialized VHH epitope recognition. All four of our top VHH candidates reduced viral titers in cotton rats when administered as VHH-Fc fusions. This effect was strongest in the lungs, where viral titers were reduced to the limit of detection and outperformed prophylactic treatment with PIA174. We also observed a decrease in viral titers in nasal turbinates, which were reduced to 2–2.8 $\log_{10}$ PFU/g, whereas PIA174 only reduced titers to 3.9 $\log_{10}$ PFU/g. While neutralizing VHHs developed against influenza virus, RSV, and SARS-CoV-2 have been reported to reduce viral replication in animal models, comparisons between VHHs and monoclonal antibodies are limited[59–61]. We note that administration of antibody PIA174 and VHH-Fc fusions at equivalent doses by mass results in fewer molecules of PIA174 (~150 kDa) than the smaller VHH-Fcs (~80 kDa). A major benefit of the small size of VHH-based therapeutics is increased tissue penetration, which may also improve protective efficacy. The increased reduction in viral titers we observed in our animal model suggests that these VHH-Fc fusions should be considered for the development of interventions to combat RV3 infection.

## Methods

All studies were conducted in compliance with local, state, and federal rules and regulations. Structural and biochemical studies performed at The University of Texas at Austin were approved by the UT Austin Institutional Biosafety Committee (protocol IBC-2023-00263). Llama immunization studies were performed under approval by the Research Ethics Committee (CER, protocol #CE-SANTE-001). Approximately one-year-old male llamas ($n = 3$) were used for the studies. Cotton rat studies were conducted under applicable laws and guidelines and after approval from Sigmovir Biosystems, Inc.'s Institutional Animal Care and Use Committee (IACUC). Inbred *S. hispidus* cotton rats were obtained from a colony maintained at Sigmovir Biosystems, Inc. Six-to-eight-week-old male cotton rats ($n = 42$) were used for the studies. Animals were housed in large polycarbonate cages and fed a standard diet of rodent chow and water ad libitum.

### Llama immunization

Three llamas were immunized subcutaneously with 100 µg of recombinant RV3 F protein on days 0 and 7, and 50 µg of recombinant RV3 preF protein on days 14, 21, 28, and 35. The RV3 preF protein was adjuvanted with Incomplete Freund's Adjuvant for all six immunizations. The F antigen contains a GCN4-trimerization domain and a D452N substitution to stabilize RV3 F in the prefusion conformation. 250 mL anticoagulated blood was collected 7 days after the final immunization for the preparation of peripheral blood lymphocytes.

### Phage library reconstruction

VHH-presenting phage libraries were generated as follows. Briefly, 40 µg of total RNA isolated from peripheral blood lymphocytes of the immunized animals was converted into cDNA through reverse transcription using random primers. The synthesized cDNA was used as a template for subsequent PCR amplification of VHH encoding gene fragments. These fragments were digested and cloned into a pDCL1 phagemid vector. The resulting vectors were transformed into electrocompetent *E. coli* TG1 cells to generate VHH libraries of 1–1.5 × 10$^8$ independent transformants.

### Expression and purification of RV3 F proteins

Transient transfection of Expi293F cells (Catalog number: A14527) in 300 mL scale was performed to obtain supernatant for purification.

Cells at a density of $2.5 \times 10^6$ vc/mL in Expi293F Expression medium [+] GlutaMAX (Gibco) were transfected using an ExpiFectamine 293 Transfection Kit (Gibco) according to the instructions of the manufacturer and cultured in 1 L vented flasks (Corning) at 37 °C, 75% humidity, 125 rpm, 8.0% CO2. Enhancers were added one day post transfection and five days post transfection the supernatants were harvested using centrifugation (10 min at 600 x g) and subsequently clarified by a 0.22 µm PES vacuum filter (Nalgene) to obtain a sterile product. Supernatant containing RV3 F was tested for binding to prefusion-specific antibody PIA174 (500 nM)[14] to ensure proper protein conformation. The sterile supernatant was stored at 4 °C until further use.

RV3 trimers containing a C-tag or His-tag were purified on an ÄKTA Avant 25 system (Cytiva) using a two-step protocol. To a 5 mL capture Select C-tag XL column (Thermo Scientific) equilibrated in PBS pH 7.4 (1x, Gibco) the 0.22 µm filtered supernatant containing C-tagged protein was applied. Subsequently, the column was washed with PBS and the proteins were eluted with 20 mM Tris, 2 M $MgCl_2$, pH 7.0. Elution fractions were 1:1 diluted with 20 mM Tris, pH 8.0 to lower the $MgCl_2$ concentration. For RV3 trimers containing a His-tag the 0.22 µm filtered supernatant was conditioned with 20 mM Tris, 500 mM NaCl, 500 mM imidazole pH 7.5 to obtain a final imidazole concentration of 35 mM before being applied to a 5 mL HisTrap HP column (Cytiva) equilibrated in 20 mM Tris, 500 mM NaCl, 35 mM imidazole, pH 7.5. Subsequently the column was washed with 20 mM Tris, 500 mM NaCl, 35 mM imidazole, pH 7.5 and the proteins were eluted with 20 mM Tris, 500 mM NaCl, 500 mM imidazole, pH 7.5. The obtained His- and C-tag elution pools were concentrated using an Amicon 50 kDa MWCO filter (Millipore) prior to size exclusion for further purification. Concentrated proteins were applied to a Superdex200 Increase 10/300GL column (Cytiva) equilibrated in 20 mM Tris, 75 mM NaCl pH 7.4. Monodisperse fractions were pooled and sterile 0.22 µm filtered to obtain a final product. Proteins were stored at 4 °C or for long term storage at −80 °C snap frozen in liquid nitrogen.

### Biotinylation

Purified protein was randomly biotinylated using the EZ-Link™ NHS-LC-LC-Biotin kit (Thermo Fisher Scientific, #21343) according to the manufacturer's instructions. The NHS-LC-LC-Biotin was dissolved to 10 mM in dimethyl sulfoxide (DMSO) prior to use. If necessary, the protein was desalted to 1x PBS, pH 7.4 (Gibco) using Zeba desalting columns 40 kDa MWCO (Thermo Fischer, #87768) prior to biotinylation to avoid buffers containing primary amines. After biotinylation, the protein was again desalted to remove excess non-reacted and hydrolyzed biotin reagent and to have the protein in the desired buffer for further experiments.

### Isolation of RV3 F-reactive VHH phages

Phages displaying RV3 F-directed VHHs were enriched after 2 rounds of biopanning under different selection conditions. In round I, all phage libraries were incubated with various concentrations (ranging from 0.1 to 50 nM) of biotinylated RV3 preF, either in solution or captured on Dynabeads™ MyOne™ Streptavidin T1 Magnetic Beads (Invitrogen, #65601). To eliminate RV3 postF- and GCN4-binding VHH domains, a 20–10,000-fold excess of RV3 postF and GCN4 peptide was added. After 2 h of incubation at room temperature, streptavidin magnetic beads were added to the samples containing biotinylated RV3 preF and all bead suspensions were subsequently washed 6 times with 1x PBS-Tween 0.05% and twice with 1x PBS. Phages bound to biotinylated RV3 preF on the beads were eluted with triethylamine solution (0.1 M) for 10 min and subsequently neutralized with 1 M Tris-HCl, pH 7.4. In round II, enriched phage outputs from round I were incubated with different concentrations of biotinylated RV3 preF (ranging from 0.01 nM to 10 nM) in the absence and presence of up to

105-fold excess of RV3 postF and GCN4 peptide. After 2 h of incubation at room temperature, streptavidin beads were added to capture the biotinylated RV3 preF from the solution and the beads were subsequently washed 6 times with 1× PBS-Tween 0.05% and twice with 1x PBS. Phages bound to biotinylated RV3 preF on the beads were eluted with 1 mg/mL Trypsin (Sigma, Cat nr. T1426-5G) for 20 min and subsequently neutralized by addition of 10 µL AEBSF (Sigma, Cat nr. A8456). Round I and II selection outputs showing high enrichment over background at low RV3 preF concentrations were used to infect *E. coli* TG1 cells. After plating the infected cells on LB Agar, single clones were picked and grown overnight in 96 deep-well plates for the preparation of glycerol stocks and VHH-containing periplasmic extracts.

### ELISA

For serum titration experiments performed on pre- and post-immune samples from the immunized llamas were analyzed for binding to RV3 pre- and postF antigens. Maxisorp 96-well microtiter plates were coated with 5 µg/mL Neutravidin (Thermo, #31000) in 1x PBS overnight at 4 °C and subsequent blocking was performed with 1% Casein/1x PBS (Sigma, # C7078-500G) for 2 h at room temperature. For antigen capture, biotinylated antigens diluted in 0.1% Casein/1x PBS to 5 nM was added to each well and incubated for 1 h at room temperature. Next, 3-fold serial dilutions (in 0.1% Casein/1x PBS) of llama pre- or post-immunization sera ranging from 1:2700-1:5314410 were added to wells and incubated at room temperature for 1 h. Bound single domain antibodies were detected by addition of 1:5000 Goat anti-llama IgG-heavy and light chain antibody-HRP (Bethyl, #A160-100P) to each well, followed by a 1 h incubation at room temperature and subsequent addition of TMB substrate (eBioscience, #00-4201-56). The reaction was stopped using 100 µL of $H_2SO_4$ (Fisher Chemical, #J/8430/15) and O.D at 450 nm was detected using a microplate spectrophotometer.

VHH-containing periplasmic extracts (P.E.) were analyzed for binding to three biotinylated antigens: RV3 preF, RV3 postF and GCN4 peptide. Capture and blocking were performed as described above. After antigen capture, five-fold dilutions of P.E. samples were added to each well and incubated for 1 h. Bound, Myc-tagged VHH domains were detected by addition of mouse anti-c-myc IgG-HRP, donkey anti-mouse IgG-HRP, and TMB substrate. O.D. at 450 nm was detected using a microplate spectrophotometer.

Purified VHHs were analyzed for binding to biotinylated RV3 preF and RV3 postF. Capture, blocking, and detection were performed as described above. Three-fold serial dilutions (500 nM–0.003 nM) of purified VHH domains or an irrelevant VHH were added to antigen-containing wells and incubated for one hour prior to detection.

### Off-rate analysis of RV3 F VHH clones using surface plasmon resonance

Biotinylated RV3 preF or postF were immobilized on a SA sensor chip (GE Healthcare, #BR-1003-98). 1:5 diluted P.E. samples in HBS-EP buffer, pH 7.4 were injected over 2 min at 30 µL/min. Dissociation was measured for 360 s. A regeneration step was performed between each cycle using 10 µL 10 mM Glycine, pH 2.5 followed by 10 µL of 10 mM NaOH/1 M NaCl. Off-rate values ($k_d$) were calculated by fitting the obtained sensorgrams to a 1:1 Langmuir dissociation model in the BIACORE T200 Evaluation software.

For VHH clones that were positive for F binding by ELISA but failed to bind RV3 F under the above experimental conditions, Biotin CAPture reagent was immobilized on the CAP sensor chip according to the manufacturer's instructions. Biotinylated RV3 preF or postF was injected at 20 nM followed by a stabilization period. 1:5 diluted P.E. samples in HBS-EP buffer, pH 7.4 were injected, and dissociation was measured for 900 s. Full regeneration with 6 M guanidine-HCl/0.25 M NaOH was performed between cycles. Off-rate values ($k_d$) were calculated by fitting the obtained sensorgrams to a 1:1 Langmuir dissociation model in the BIACORE T200 Evaluation software.

## Expression and purification of RV3 F VHHs

A selection of 39 RV3 preF specific VHH domains were cloned into a pCB4 bacterial expression vector for production and purification. VHH gene sequences cloned into the expression vector were confirmed by sequencing. *E. coli* cells transformed with VHH-expressing pCB4 vectors were grown in 250 mL for protein production. Periplasmic extracts prepared from the harvested cell pellets were applied to a HisTrap HP 5 mL IMAC column (GE Healthcare, #17-5248-08) for purification of the His-tagged VHH domains. The eluate was buffer exchanged to 1× PBS using a HiTrap Desalting column (GE Healthcare, #17-1408-01) and concentrated using a 3 kDa MWCO spin concentrator (Amicon, #UFC900324). Protein concentration was determined by measuring the absorbance at 280 nm using a micro-volume spectrophotometer. Final yields of purified VHH domains were between 1.5 and 6.9 mg. SDS-PAGE analysis showed that all VHH domains were highly pure and migrated at the expected molecular weight (between 14-17 kDa).

## In vitro neutralization of recombinant RV1-eGFP, PIV2-eGFP, and RV3-eGFP virus

10 μL of each VHH or VHH-Fc were added to wells of a black 384-well clear-bottom microtiter plate in a 4-fold serial dilution. Next, 20 μL of LLC-MK2 (ATCC, CCL-7) cells at a density of $1.5 \times 10^5$ cells/mL were added (i.e. 3000 cells/well) followed by the addition of 10 μL of either RV1-eGFP, PIV2-eGFP, or RV3-eGFP virus (ViraTree, #P323) at a multiplicity of infection (MOI) = 0.01. For assays including RV1-eGFP or PIV2-eGFP, trypsin was added to a final concentration of 1 μg/mL. Cell controls (no virus added) and virus controls (cells infected with virus but no VHH added) were included on each plate. After two days of incubation at 37 °C and 5% $CO_2$, viral replication was quantified by measuring eGFP expression using a Fluoroscan instrument. $IC_{50}$ values were calculated from fitting four-parameter dose-response curves in GraphPad Prism v10.2.1 and are reported as geometric means of biological replicates in Supplementary Table 1, Supplementary Fig. 5b.

## Surface plasmon resonance

Biotin CAPture reagent was injected and flowed over a CAP sensor chip (GE Healthcare, #28920234) for 300 s using a BIACORE T200 (GE Healthcare). 20 nM of biotinylated RV3 preF or post F were injected at a flow rate of 20 μL/min for 300 s followed by a stabilization period of 60 s. Next, purified VHH 1A03, 1D10, 1H09, 2H02, 2E10, 3B04, 4C03, or 4C06 was injected in a 1.5-fold dilution ranging from 50 nM–4.39 nM. $K_D$ values were calculated in the BIACORE software using the 1:1 interaction Langmuir dissociation model.

## Epitope mapping by cross-competition analysis

Competition binning was performed using an Octet RED HTX system at 25 °C, with a shaking speed of 1000 rpm using streptavidin (SA) biosensors (FortéBio). Biosensors were first equilibrated in 1× kinetic buffer (FortéBio) for 10 min before an initial 60 s baseline "sensor check" by dipping the biosensors into wells containing 1× kinetic buffer. Next, biosensors were loaded with antigen by immersion in wells containing 10 μg/mL of RV3 preF for 600 s followed by immersion in running buffer for 60 s to get a baseline. Then antigen-loaded tips were immersed into wells containing VHH (15 μg/mL), or running buffer for 600 s, followed by another 60 s baseline step of buffer-only wells. Finally, saturated antigen was immersed into 15 μg/mL of secondary VHH or running buffer for 600 s and the final response signals were observed to determine competition with the primary single domain antibody. FortéBio software Octet Data Analysis High Throughput v10.0 was used to generate an epitope binning competition matrix. Data were then transferred to GraphPad Prism v10.2.1 for data visualization.

## RV3 F + VHH complex formation and purification

To form a stable complex for structural studies, 0.75 mg of RV3 F protein was mixed with 1.2X molar excess each of VHH 4C06 and either 4C03 or 1D10 and incubated on ice for one hour to promote binding saturation. The complex was then separated from individual components by size-exclusion chromatography using a Sepharose 6 10/300 column (GE Healthcare) in buffer containing 2 mM Tris, pH 8.0, 200 mM NaCl and 0.02% $NaN_3$. Fractions containing complex were combined and concentrated using an Amicon 30 kDa MWCO spin concentrator and flash frozen.

## Cryo-EM sample prep and data collection

RV3 F-VHH complexes were thawed and diluted to 2 mg/mL in buffer containing 2 mM Tris pH 7.5, 200 mM NaCl, 0.02% $NaN_3$ + 0.01% amphipol. 4 μL of sample was applied to Au-300 1.2/1.3 grids (UltrAuFoil) that had been plasma cleaned for 4 minutes using a 4:1 ratio of $O_2$:$H_2$ in a Solarus 950 plasma cleaner (Gatan). Using a Vitrobot Mark IV (Thermo Fisher), after a 10 s wait a blot force of −1 was applied for 6 s to blot away excess liquid before plunge-freezing into liquid ethane. Samples were blotted in 100% humidity at 22 °C. For the RV3 F-4C03-4C06 complex, 1842 movies were collected from a single grid using a Glacios TEM (Thermo Fisher) equipped with a Falcon 4 detector (Thermo Fisher). For the RV3 F-1D10-4C06 complex, 1,993 movies were collected on a single grid as above, with the stage tilted to 30°. All movies were collected using SerialEM v4.0.10 automation software[62]. Particles were imaged at a calibrated magnification of 0.94 Å/pixel, with a dose of 3.6 eps for 14 s for a total dose of 50 e/Å². Additional details about data collection parameters can be found in Supplementary Table 4.

## Cryo-EM processing and structure building

For the RV3 F-4C03-4C06 complex, motion correction, CTF estimation, particle picking, and preliminary 2D classification were performed using cryoSPARC v4.0.0 live processing[63]. The final iteration of 2D class averaging distributed 580,195 particles into 80 classes using an uncertainty factor of 3. From that, 532,816 particles were selected and a subset of 100,000 particles were used to perform an ab initio reconstruction with four classes followed by heterogeneous refinement of those four classes using all selected particles. Particles from the highest quality class were used for homogenous refinement of the best volume with no applied symmetry. After iterative homogenous refinements, a final refinement was performed with applied C3 symmetry and with optimized per-particle defocus and per-group CTF parameters[64]. To improve map quality, the refinement volumes were processed using the DeepEMhancer tool via COSMIC² science gateway[65]. An initial model was generated by docking PDBID: 6MJZ into the refined volume via ChimeraX v1.7[14,66]. The structure was iteratively refined and completed using a combination of Phenix v1.18.2, Coot v0.9.2, and ISOLDE v1.7[67–69].

For the RV3 F-4C06-1D10 complex, motion correction, CTF estimation, particle picking, and 2D classification were performed using cryoSPARC v4.0.3 live processing[63]. 2D class averaging distributed 1,669,965 particles into 50 classes. From that, 972,572 particles were selected, and a subset of 100,000 particles was used to perform an ab initio reconstruction with one class. A subset of 150,000 particles from the 697,393 excluded particles was used to perform an ab initio reconstruction with three classes. Heterogenous refinement of those four classes was performed using the selected particles. Particles from the main class were passed through a class probability filter with a 3D class threshold of 0.8, resulting in 707,800 particles. These particles were used for homogenous refinement of the main volume from the heterogeneous refinement with no applied symmetry. Two iterative non-uniform refinements were performed, first with no applied symmetry, then applied C3 symmetry. Next, to help sort particle heterogeneity, a subset of 200,000 particles was used for an ab initio reconstruction with three classes and a class similarity value of 0.8, followed by heterogeneous refinement of all classes. The volume that appeared to be the most closed at the apex of RV3 F was selected for

further refinement. The 238,718 particles from that class were sorted for high nanobody occupancy using 3D classification, resulting in 56,650 final particles. These particles were used for a non-uniform refinement of the best 3D classification volume with applied C3 symmetry. Map quality of this volume was improved using the DeepEMhancer tool, and an initial model was generated using the Model Angelo tool, both via the COSMIC[2] science gateway[65,70]. The structure was iteratively refined and completed using a combination of PyMOL v4.6.0 (Schrodinger), Phenix v1.18.2, Coot v0.9.2 and ISOLDE v1.7[67–69]. Structure validations and full cryo-EM processing workflows can be found in Supplementary Figs. 6–8.

## TMPRSS2 production and purification

The TMPRSS2 protein used in the RV3 F cleavage inhibition assay was generously provided by the Structural Genomics Consortium, at the University of Toronto. Protein was produced and characterized as described in ref. 34.

Prior to its use in enzymatic assays, TMPRSS2 was dialyzed against 25 mM Tris-Cl pH 8, 75 mM NaCl, 2 mM $CaCl_2$ using a 10 kDa MWCO dialysis cassette (Thermo Scientific) to remove the benzamidine inhibitor included for stable storage of the protein.

## Cleavage inhibition assay

To determine whether 4C06 binding could inhibit cleavage by activating proteases, RV3 F-His ($3.6\,\mu M$) was incubated with or without $5\,\mu M$ of 4C06 in reaction buffer (25 mM Tris-Cl pH 8, 75 mM NaCl, 2 mM $CaCl_2$) for 10 min before adding 100 nM Trypsin (Sigma-Aldrich) or 100 nM TMPRSS2[34]. 100 nM of HRV 3C protease was added as a negative control. All reactions were incubated at a final volume of $50\,\mu L$. After 30 min, reactions were quenched by adding nafamostat (MedChemExpress) to a final concentration of $1\,\mu M$. Samples were combined with SDS loading dye containing BME and boiled prior to analysis by SDS-PAGE. An uncropped image of the gel shown in Fig. 4c is available in the provided Source Data file.

## Generation of Fc-fusion constructs

VHH-Fc fusion constructs were generated for VHH 4C03, 4C06, 1H09, and 1D10. The VHH domains were fused to the Fc domain of human IgG1 consisting of a hinge region followed by domains CH2 and CH3. The full-length VHH-Fc genes were directly synthesized and ligated into a eukaryotic expression vector. Complete amino acid sequences of the Fc fusion constructs are shown in Supplementary Table 5. Fc-fusion constructs were expressed in HEK293E-253 cells. Six days post-transfection, conditioned medium containing recombinant dimeric Fc molecules was harvested by low-speed centrifugation followed by high-speed centrifugation. All constructs were purified using MabSelectSure LX Sepharose resins and eluted using 20 mM citrate, 150 mM NaCl, pH 3.0 in fractions containing 1 M $K_2HPO_4/KH_2PO_4$, pH 8.0 buffer for neutralization to pH 7.0. Possible aggregates and impurities were removed by preparative gel filtration using a Superdex200 Increase 26/40 that was equilibrated in PBS. Purified proteins were further analyzed by SDS-PAGE and concentrated using Amicon Ultra 30 kDa centrifugal filters (Supplementary Fig. 5a).

## Cotton rat infection experiments

6–8-week-old male *Sigmodon hispidus* cotton rats (Source: Sigmovir Biosystems, Inc., Rockville MD) ($n = 6$) were dosed via intramuscular injection with 4C03-Fc, 4C06-Fc, 1D10-Fc, 1H09-Fc, or antibody PIA174 at a single dose of 15 mg/kg. Another group of 6 cotton rats receiving buffer solution only served as a vehicle control group. The last group of 3 cotton rats were not infected with virus but dosed with PBS and served as the uninfected control group. 24 h post-administration, cotton rats were challenged intranasally with $10^5$ PFU RV3 C243 strain (ATCC, Manassas, VA). To determine lung and nasal viral titers, cotton rat lung and nose tissues were homogenized in 3 mL of

HBSS supplemented with 10% sucrose phosphate glutamate media (SPG) in Bertin auto homogenizer. The mixture was then spun at 2000 rpm for 10 min at 4 °C. Clarified lung homogenates were diluted 1:10 and 1:100 in EMEM. Confluent Hep-2 monolayers in 24-well plates were infected in duplicate with 50 µL of sample per well starting with undiluted (neat) samples, followed by diluted homogenates. After one hour incubation at 37 °C and 5% CO2, wells were overlaid with 0.75% methylcellulose medium, and plates returned to the incubator. After 4 days, the overlay was removed and cells were fixed with 0.1% crystal violet stain for one hour, then rinsed and air-dried. Plaques were counted and viral titers were expressed as plaque forming units per gram of tissue. Viral titers for each group were calculated as the mean values with standard deviation for all animals in that group.

## Data processing and statistical analysis

Statistical analyses were performed using Prism v10.2.1 (GraphPad Software). Prism v10.2.1 was also used to plot the data. Information about the statistical tests performed can be found in the figure legends.

## Reporting summary

Further information on research design is available in the Nature Portfolio Reporting Summary linked to this article.

## Data availability

The amino acid sequences of the 13 neutralizing VHHs are available in Supplementary Table 2 and the sequences for the VHH-Fc constructs are in Supplementary Table 5. Structural models are deposited in the protein data bank (PDB, https://www.rcsb.org/). The PDB IDs are: 8V5K and 8V62. Cryo-EM maps are deposited in the EM database (https://www.emdataresource.org/). The maps are publicly available and can be found using the following EMDB IDs: 42983 and 42987. All data supporting the findings of this study are within the article and its Supplementary Information. Source data are provided with this paper.

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

## Acknowledgements

Research performed at Janssen Infectious Diseases and Vaccines and Janssen Vaccines & Prevention BV was funded by Janssen Pharmaceuticals. Research performed in the McLellan Lab was funded in part by a sponsored research agreement with Janssen Pharmaceuticals and by Welch Foundation grant no. F-0003-19620604 (J.S.M.). The authors thank Dr. Axel F. Brilot at the Sauer Structural Biology Laboratory at the University of Texas at Austin for technical assistance with microscope operation. We also thank the Structural Genomics Consortium, at the University of Toronto, for providing the TMPRSS2 protein used in this study. We thank Alison Lee for help with statistical analyses and Ling Zhou for the cotton rat drawing included in Fig. 6b. Figures 1b and 6b were created with BioRender.com. We would like to thank Fernando Martins and the team at FairJourney Biologics for their assistance in identifying and analyzing the RV3 F VHHs, and Fin Milder for technical assistance and fruitful discussions.

## Author contributions

Conceptualization: M.J.G.B., J.P.M.L., J.A.K., and J.S.M. Investigation: N.V.J., R.C.v.S., M.J.G.B., A.R.R., D.v.O., L.L., and J.P.M.L. Cryo-EM characterization: N.V.J., A.R.R., and J.S.M. Manuscript writing (original draft): N.V.J., R.C.v.S., M.J.G.B., A.R.R., and J.P.M.L. Manuscript editing: N.V.J., R.C.v.S., M.J.G.B., A.R.R., J.A.K., J.P.M.L., and J.S.M.

## Competing interests

J.P.M.L. and M.J.G.B. are co-inventors on related vaccine patents. D.v.O. and L.L. are employees of Johnson & Johnson. The remaining authors declare no competing interests.
