## [Peer Review File · Nature Communications]

Structural basis for potent neutralization of human
respirovirus type 3 by protective single-domain camelid
antibodiesREVIEWER COMMENTS

Reviewer #1 (Remarks to the Author):

The paper by Johnson et al. makes significant contributions in several respects. The work shows that camelid heavy-chain-only antibody fragments (VHHs) can be generated using pre-fusion F of human parainfluenza virus type 3 (RV3 / hPIV3) to efficiently neutralize RV3; that the VHHs target distinct antigenic sites, clearly delineated at high resolution by cryo-EM, different from those targeted by previously characterized antibodies; and that they likely work by several distinct mechanisms.

The paper is extremely well written and clear, and is very interesting to read. The figures are absolutely clear to work through, the methods and supplemental information are clear and well presented, and the paper is extraordinarily well referenced.

Approaches to generate neutralizing antibodies for this pathogen via scalable, rigorous methods are very important. This is a virus that causes a significant burden of disease. This paper shows that a rigorous pipeline to generate stabilized fusion protein antigens, followed by generation of antibodies, structural determination of specific binding sites, epitope mapping, and assessment of neutralization potency, is eminently feasible. The epitopes identified in the work are important for future development of prophylactics and therapeutics and also for mechanistic studies.

Lines 111-113. It would be of interest to know more about the down-selection – how the 39 lead VHH candidates were arrived at (“based on their F-binding profiles”).

Lines 173-174. The speculation, based on the Ab-F structure, that Ab binding locks the prefusion state is important. Here and onward, there are several such mechanistic hypotheses that would be of real interest to test experimentally. The mechanistic study is not the focus of the current paper and not necessary to it, and would be a topic for future study.

Another example is in lines 214-216 where the structure suggests that an Ab is binding to a domain that modulates HN-F interaction. It would be exciting in future work to explore this. The authors note in the Discussion that “direct experimental evidence is required to confirm” these mechanisms, and one of the exciting aspects of this paper is the set of hypotheses that it opens up for exploration.

Reviewer #2 (Remarks to the Author):

Several VHHs (also known as sdAbs or Nanobodies) with specificity against F (fusion) protein of respirovirus 3 were retrieved from an immune library (obtained after immunising 3 llamas with

perfusion-stabilized F antigen. A few lead sdAb candidates with potential virus neutralisation potential were identified and further investigated by cryo-EM of the sdAb-PreF complex to understand the neutralisation mechanism. Finally, in a cotton rat model, the VHH-Fc fusion prophylactically administered, appeared to prevent virus proliferation.

This work is of significant interest as the respirovirus 3 is still causing major health problems (causing pneumonia and many hospitalisations) in young children. In addition, the classical antibodies seem to be less protective as they tend to be directed against non-essential epitopes of the virus. Here the VHHs seem to prefer - and support the hypothesis - the recognition of alternative epitopes.

Unfortunately the VHHs recognise only the respirovirus3 F protein and not the corresponding protein of closely related paramyxoviruses, which limits its applicability as a prophylactic therapeutic.

The work is solid and presented in a logical order. The methodology is described with sufficient detail. The data analysis seems to be performed according to standards without obvious flaws. However, the authors are invited to amend or clarify a few things:

- Figure 2C is hard to understand. For example: what is the explanation of the observation of 'intermediate binders' (those with signal between 0.33-0.66 nm??)
- More importantly, the authors should realign the numbering to the amino acid sequences in Fig 2D. For VHHs it is best to use the IMGT numbering AND ALSO the CDR1, CDR2 and CDR3 annotations. (For example a sequence alignment among CDR3s with a different length should not be done) The CDR2 is far too extended, the boundaries of the CDR1 are not properly made it should start at G27, and end before the M/L39 (IMGT numbering)
- The low k_d values of the 4 binders is due to avidity affects during the washings. The antigen is trivalent and the VHH is monovalent, so during the washing you will have rebinding of the released VHH to one of the two free epitopes on the same trimeric molecule. In a 1 to 1 Ag/VHH reaction it is very rare to have a k_d below $10E-3$ s⁻¹.
- Figure 3B: please use different colours for the CDR1, CDR2 and CDR3 (also make the CDR3 according to IMGT and not to Kabat as this is too extended. Evidently, use the same CDR colouring in subsequent panels (and figures)
- Figure 4B If you would use the CDR classification of IMGT you will avoid the FR1 interaction with Ag. Ser27 and Ser29 in your numbering will be part of the antigen-binding loop of a VHH.
- The interaction with Asn76 and Asn73 is not that extraordinary. It is often named the 4th hyper variable loop that interacts occasionally with (large) antigens.
- Please renumber the Aa from the (sequential) crystallographic numbering to the IMGT numbering which makes more sense in biology (molecular immunology)

Reviewer #3 (Remarks to the Author):

Johnson and colleagues successfully identified numerous single-domain camelid antibodies from llamas immunized with RV3 (hPIV3) F. Out of 500 clones, 13 demonstrated neutralizing activity, and

a comprehensive characterization was conducted on four of them. Overall, this work shares important results, is well written and should be of interest to Nat Commun readers.

1. How is it possible to isolate postF-specific antibodies from preF-stabilized molecules? Could this imply that the immunogen used in the immunization process is not yet fully pre-fusion stabilized? The paper lacks physical and biochemical characterization of the immunogens. For instance, the author employed PIA174 in the challenge study, but there is no indication that PIA174 was used for the characterization of immunogens.

2. While the paper and its figures predominantly emphasize the structure and neutralization of the four VHH, it is essential to incorporate data related to other neutralizing VHH antibodies. This should include information such as their sequences, binding characteristics, competition data, and neutralization titers.

3. Fig2 c, lack of full of binning data.

4. It is interesting that the selected VHH antibodies exhibit enhanced protection in the challenge experiment compared to the human neutralizing antibody PIA174. To substantiate this finding, it is crucial to assess the serum levels of antibodies and establish a correlation with the observed protection. Without conducting thorough serum analysis, drawing the conclusion that VHH antibodies outperform the human antibody may lack persuasiveness. The intramuscular injection of VHH with Fc (molecular weight ~100 kDa) and the human antibody (molecular weight ~150 kDa) in equivalent amounts suggests that the human antibody is dosed in a molar quantity less than the single chain. This also need to be discussed in the manuscript.

5. Single-domain antibodies derived from VHH are widely used as building blocks in antibody engineering. The authors have successfully pinpointed VHH antibodies that specifically target various antigenic sites. To enhance the impact of their research, the authors could consider re-engineering VHH antibodies using structural information. For instance, creating tandem or bi-specific antibodies could potentially amplify their neutralization potency through synergistic effects.

REVIEWER COMMENTS

Reviewer #1 (Remarks to the Author):

The paper by Johnson et al. makes significant contributions in several respects. The work shows that camelid heavy-chain-only antibody fragments (VHHs) can be generated using pre-fusion F of human parainfluenza virus type 3 (RV3 / hPIV3) to efficiently neutralize RV3; that the VHHs target distinct antigenic sites, clearly delineated at high resolution by cryo-EM, different from those targeted by previously characterized antibodies; and that they likely work by several distinct mechanisms.

The paper is extremely well written and clear, and is very interesting to read. The figures are absolutely clear to work through, the methods and supplemental information are clear and well presented, and the paper is extraordinarily well referenced.

Approaches to generate neutralizing antibodies for this pathogen via scalable, rigorous methods are very important. This is a virus that causes a significant burden of disease. This paper shows that a rigorous pipeline to generate stabilized fusion protein antigens, followed by generation of antibodies, structural determination of specific binding sites, epitope mapping, and assessment of neutralization potency, is eminently feasible. The epitopes identified in the work are important for future development of prophylactics and therapeutics and also for mechanistic studies.

Lines 111-113. It would be of interest to know more about the down-selection – how the 39 lead VHH candidates were arrived at (“based on their F-binding profiles”).

Response: We agree that additional details about the down-selection process would be of interest to readers. The following criteria were used to select the panel of 39 VHH lead candidates:

- **Primary hits were defined as all VHH clones that showed strong binding to PIV3 preF in ELISA**
- **All primary hits for which DNA sequencing failed were deselected**
- **All primary hits with SPR off-rates higher than $5E-03$ s⁻¹ were deselected**
- **Remaining primary hits were categorized in VHH clusters based on CDR3 sequence homology: from each cluster 1 representative with the slowest off-rate (strongest binder) was selected for further characterization.**

To summarize this in the text, we modified the previous lines 111–113 to read: “Clones that bound preF with slow off-rates (less than 5×10^{-3} s⁻¹) were assigned to a VHH cluster based on CDR3 sequence homology. The strongest binder from

each cluster was selected to generate a panel of 39 lead VHH candidates to recombinantly express and purify for further characterization.”

Lines 173-174. The speculation, based on the Ab-F structure, that Ab binding locks the prefusion state is important. Here and onward, there are several such mechanistic hypotheses that would be of real interest to test experimentally. The mechanistic study is not the focus of the current paper and not necessary to it, and would be a topic for future study.

Another example is in lines 214-216 where the structure suggests that an Ab is binding to a domain that modulates HN-F interaction. It would be exciting in future work to explore this. The authors note in the Discussion that “direct experimental evidence is required to confirm” these mechanisms, and one of the exciting aspects of this paper is the set of hypotheses that it opens up for exploration.

Response: We agree with the reviewer that the suggested experiments are interesting, and we are working collaboratively to address these hypotheses.

Reviewer #2 (Remarks to the Author):

Several VHHs (also known as sdAbs or Nanobodies) with specificity against F (fusion) protein of respirovirus 3 were retrieved from an immune library (obtained after immunising 3 llamas with prefusion-stabilized F antigen). A few lead sdAb candidates with potential virus neutralisation potential were identified and further investigated by cryo-EM of the sdAb-PreF complex to understand the neutralisation mechanism. Finally, in a cotton rat model, the VHH-Fc fusion prophylactically administered, appeared to be prevent virus proliferation.

This work is of significant interest as the respirovirus 3 is still causing major health problems (causing pneumonia and many hospitalisations) in young children. In addition, the classical antibodies seem to be less protective as they tend to be directed against non-essential epitopes of the virus. Here the VHHs seem to prefer - and support the hypothesis - the recognition of alternative epitopes.

Unfortunately the VHHs recognise only the respirovirus3 F protein and not the corresponding protein of closely related paramyxoviruses, which limits its applicability as a prophylactic therapeutic.

The work is solid and presented in a logical order. The methodology is described with sufficient detail. The data analysis seems to be performed according to standards without obvious flaws. However, the authors are invited to amend or clarify a few things:

1. Figure 2C is hard to understand. For example: what is the explanation of the observation of 'intermediate binders' (those with signal between 0.33-0.66 nm??)

Response: Thank you for bringing this to our attention. To add clarity to the figure, we now indicate the directionality of the competition experiment (first bound VHH, second bound VHH) and added colored boxes (consistent with the colors assigned to VHHs throughout the figures) to the heatmap to emphasize the different epitope bins, including one to indicate the overlapping epitope bound by VHH 1H09.

Experiments with intermediate values (0.33-0.66 nm) can be observed in several situations. Binding of the second VHH to RV3 preF may be partially occluded by the first VHH to an extent that is not consistent with full epitope overlap, resulting in reduced binding that remains well above the buffer-only signal (0-0.2 nm). Partial competition signals may also result from competition with a first VHH that has a sufficient off-rate to at least partially dissociate during the second binding event, allowing the second VHH to bind a shared epitope. Because of this ambiguity, intermediate binding is not considered for assignment of epitope bin. By contrast, second VHHs that bind with values of 0-0.33 nm are within or just above the buffer-only signal, indicating full or nearly full competition with the first VHH bound.

To clarify, we have added the following sentence to the Results section, lines 135-140: "Competition between VHHs was defined by the level of response (nm) observed for binding of the second VHH. VHHs that competed for binding to an overlapping epitope (0 – 0.33 nm) were assigned to the same epitope bin, while those that bound strongly to the complex (0.66 – 2 nm) were assigned to a separate bin. Intermediate response values (0.33 – 0.66 nm) can indicate minor epitope overlap or exposure of a shared epitope through partial dissociation of the first VHH and were not considered for bin assignment."

2. More importantly, the authors should realign the numbering to the amino acid sequences in Fig 2D. For VHHs it is best to use the IMGT numbering AND ALSO the CDR1, CDR2 and CDR3 annotations. (For example a sequence alignment among CDR3s with a different length should not be done) The CDR2 is far too extended, the boundaries of the CDR1 are not properly made it should start at G27, and end before the M/L39 (IMGT numbering)

Response: Thank you for this comment. This is an important correction for consistency in the field and we appreciate you bringing this to our attention. We included a new alignment in figure 2D that includes the sequential amino acid number, the IMGT number, and boxes to indicate the CDRs for each VHH. The VHHs are aligned using VHH 4C03, which has the longest CDR1, CDR2,

and CDR3 of the four VHHs included. As such, dashes indicate missing positions within the CDRs of the remaining three VHHs and the numbering (sequential and IMGT) are preserved for all of them.

3. The low k_d values of the 4 binders is due to avidity affects during the washings. The antigen is trivalent and the VHH is monovalent, so during the washing you will have rebinding of the released VHH to one of the two free epitopes on the same trimeric molecule. In a 1 to 1 Ag/VHH reaction it is very rare to have a k_d below $10E-3 s^{-1}$.

Response: Thank you for this comment. Respectfully, we disagree with the assessment that the determined K_D values are low due to avidity. Our experimental setup includes the trivalent antigen as the immobilized ligand and the monovalent VHH as the flowing analyte. Under these conditions using SPR, the valency of the ligand does not affect the response detected for binding and release of analyte. Artificially slow off-rates due to avidity occur when the analyte is multivalent, as the analyte can then bind multiple epitopes simultaneously, and dissociation would only be detected when all binding sites have been released.

The K_D values we report are consistent with previous observations of VHH binding to viral glycoproteins reported by us and others. For example, a panel of over 100 VHHs characterized for binding to the SARS-CoV-2 RBD, monomeric S1, or trimeric S2 displayed off-rates ranging from $10E-3 - 10E-6 s^{-1}$ (Mast et al., 2021). Additionally, we previously reported two VHHs with off-rates $\sim 10E-5 s^{-1}$ that bound to RSV preF, which is more closely related to RV3 F and is a similar size (Rossey et al., 2017).

4. Figure 3B: please use different colours for the CDR1, CDR2 and CDR3 (also make the CDR3 according to IMGT and not to Kabat as this is too extended. Evidently, use the same CDR colouring in subsequent panels (and figures)

Response: Thank you for the suggestion. We updated figure 3B to depict the CDRs with the proper IMGT numbering. Our goal with this figure panel is to illustrate the angle of approach with which the VHH binds and highlight the curvature adopted by CDR3 in the interaction. We attempted several alternative color combinations and representations of the CDRs but were not satisfied that any of them added clarity and elected to maintain the original coloring scheme.

5. Figure 4B If you would use the CDR classification of IMGT you will avoid the FR1 interaction with Ag. Ser27 and Ser29 in your numbering will be part of the antigen-binding loop of a VHH.

Response: We applied the updated IMGT numbering to figure 4B and all the other figures where the CDR numbering is included. IMGT numbers have also been included in the revised text.

6. The interaction with Asn76 and Asn73 is not that extraordinary. It is often named the 4th hyper variable loop that interacts occasionally with (large) antigens.

Response: Thank you for informing us of this interesting VHH characteristic. The original text stated:

“Interestingly, the interweaving nature of the interaction involves framework residues, with FR1/CDR1, CDR2, and FR3 each forming loops that surround the FP at the F1/F2 protease recognition site between Arg109 and Phe110”

This has now been amended, with proper IMGT numbering, to say:

“The interweaving nature of the interaction involves residues 80-85 within framework region 3 (FR3), which is sometimes referred to as a fourth CDR (Fanning et al., 2011). CDR1, CDR2, and the FR3 loop surround the FP at the F1/F2 protease recognition site between Arg109 and Phe110”

7. Please renumber the Aa from the (sequential) crystallographic numbering to the IMGT numbering which makes more sense in biology (molecular immunology)

Response: IMGT numbering has been updated for the VHHs throughout the manuscript. In addition, the PDB submissions for the two structures within the manuscript have been updated to include the IMGT numbering for each VHH chain.

Reviewer #3 (Remarks to the Author):

Johnson and colleagues successfully identified numerous single-domain camelid antibodies from llamas immunized with RV3 (hPIV3) F. Out of 500 clones, 13 demonstrated neutralizing activity, and a comprehensive characterization was conducted on four of them. Overall, this work shares important results, is well written and should be of interest to Nat Commun readers.

1. How is it possible to isolate postF-specific antibodies from preF-stabilized molecules? Could this imply that the immunogen used in the immunization process is not yet fully pre-fusion stabilized? The paper lacks physical and biochemical characterization of the immunogens. For instance, the author

employed PIA174 in the challenge study, but there is no indication that PIA174 was used for the characterization of immunogens.

Response: We agree that isolating postF-specific antibodies after immunization with a preF-stabilized immunogen is counterintuitive. Indeed, we believe this is due to the limited stabilization approach that was applied, and destabilization of a portion of the immunogen after immunization. The preF immunogen used was only equipped with a GCN4 trimerization domain and a stabilizing D452N substitution at the 3-fold axis above HRB designed to limit electrostatic repulsion. The D452N substitution is akin to the D486N substitution that we previously reported for RSV F (Krarup et al, 2015).

Regarding the biochemical characterization of the immunogen, we have now included PIA174 binding data, as measured by BLI, as supplementary figure 2b.

2. While the paper and its figures predominantly emphasize the structure and neutralization of the four VHH, it is essential to incorporate data related to other neutralizing VHH antibodies. This should include information such as their sequences, binding characteristics, competition data, and neutralization titers.

Response: Thank you for this suggestion. We have added the sequences for the 13 lead VHH candidates to supplementary table 2 and added EC50 values for pre- and postF binding to supplementary table 3. Additionally, neutralization for all 39 initial lead VHH candidates was tested and IC50 values can be found in supplementary table 1.

3. Fig2 c, lack of full of binning data.

Response: Binning data was collected for the 13 lead VHH candidates. We have now included the numerical response values as a table within supplementary figure 4a.

4. It is interesting that the selected VHH antibodies exhibit enhanced protection in the challenge experiment compared to the human neutralizing antibody PIA174. To substantiate this finding, it is crucial to assess the serum levels of antibodies and establish a correlation with the observed protection. Without conducting thorough serum analysis, drawing the conclusion that VHH antibodies outperform the human antibody may lack persuasiveness. The intramuscular injection of VHH with Fc (molecular weight ~100 kDa) and the human antibody (molecular weight ~150 kDa) in equivalent amounts suggests that the human antibody is

dosed in a molar quantity less than the single chain. This also need to be discussed in the manuscript.

Response: We agree that further analyses of serum antibody titers will be needed to obtain correlates of protection, however we do not have sufficient quantities of the serum samples to perform these analyses. In lieu of these experiments, we have added a note to the manuscript stating that because the same mass of antibodies was administered, fewer molecules of PIA174 were administered than the VHH-Fc given the lower molecular weight of the VHH-Fc. Lines 335-337: “We note that administration of antibody PIA174 and VHH-Fc fusions at equivalent doses by mass results in fewer molecules of PIA174 (~150 kDa) than the smaller VHH-Fcs (~80 kDa).”

5. Single-domain antibodies derived from VHH are widely used as building blocks in antibody engineering. The authors have successfully pinpointed VHH antibodies that specifically target various antigenic sites. To enhance the impact of their research, the authors could consider re-engineering VHH antibodies using structural information. For instance, creating tandem or bi-specific antibodies could potentially amplify their neutralization potency through synergistic effects.

Response: This is a great suggestion that is beyond the scope of our current work. Given the spatial relationship between the epitopes targeted by the three VHHs that were structurally characterized, it would be interesting and feasible to engineer bi-specific molecules that target two of these epitopes simultaneously.

REVIEWERS' COMMENTS

Reviewer #1 (Remarks to the Author):

In this revised manuscript the authors have fully addressed my concerns. In addition, the paper has been improved in several aspects by responding to the comments of the review panel. I have no further concerns about this excellent paper, and I think the paper will add substantively to the respirovirus field.

Reviewer #2 (Remarks to the Author):

Again, this reviewer is impressed by the work. Each time I'm reading the manuscript, I'm discovering new details, that were overlooked on earlier readings. The experiments are carefully designed, presented in a logical order (discovery of VHH, binning, affinity, structural details of epitope:paratope interaction, engineering (reconstituted) in HCAs, testing the infection/neutralisation effect in cotton rat model).

The authors have shown their willingness to improve their report based on the comments provided by the 3 reviewers.

Reviewer #3 (Remarks to the Author):

The authors have addressed all issues raised in my review of the original version.